# Failed Back Surgery Syndrome: No Longer a Surgeon’s Defeat—A Narrative Review

**DOI:** 10.3390/medicina59071255

**Published:** 2023-07-06

**Authors:** Grzegorz Miękisiak

**Affiliations:** 1Institute of Medicine, University of Opole, 45-040 Opole, Poland; gmiekisiak@gmail.com; 2Vratislavia Medica Hospital, 51-134 Wrocław, Poland

**Keywords:** Failed Back Surgery Syndrome, Persistent Spinal Pain Syndrome, spinal surgery, chronic back pain, neuropathic pain

## Abstract

The introduction of the term Persistent Spinal Pain Syndrome (PSPS-T1/2), replacing the older term Failed Back Surgery Syndrome (FBSS), has significantly influenced our approach to diagnosing and treating post-surgical spinal pain. This comprehensive review discusses this change and its effects on patient care. Various diagnostic methods are employed to elucidate the underlying causes of back pain, and this information is critical in guiding treatment decisions. The management of PSPS-T1/2 involves both causative treatments, which directly address the root cause of pain, and symptomatic treatments, which focus on managing the symptoms of pain and improving overall function. The importance of a multidisciplinary and holistic approach is emphasized in the treatment of PSPS-T1/2. This approach is patient-centered and treatment plans are customized to individual patient needs and circumstances. The review concludes with a reflection on the impact of the new PSPS nomenclature on the perception and management of post-surgical spinal pain.

## 1. Introduction

Low back pain (LBP) is one of the leading causes of disability worldwide. The age-standardized point prevalence in 2017 was estimated to be 7.5% [1] and the lifetime prevalence can be as high as 65–80% [2]. In 2015, low back and neck pain was considered the fourth leading cause of disability-adjusted life years (DALYs) [3]. This entity has several distinctive features, making it elusive to establish rational principles for treatment. While most of the episodes of LBP are benign and short-lasting, there is a very high proportion of cases, as high as 95%, where no specific pathoanatomical pathology can be identified [4]: this predominant form is called a non-specific LBP. Among patients presenting to the emergency department, the second most common pathology is a vertebral fracture (up to 7.2%) followed by cancer and infectious disorders [5]. On the other hand, about 40–50% of chronic LBP treated in specialized clinics is alleged to be of discogenic origin [6]. Despite considerable efforts by various individuals and agencies, LBP poses tremendous challenges for healthcare systems. These include the financial interests of pharmaceutical companies, outdated systems favoring medical care over patients’ self-management [7] and obsolete fee-for-service paradigms [8].

## 2. Proposed New Nomenclature

Failed Back Surgery Syndrome (FBSS) is a controversial term covering a broad range of lumbar pain of unknown origin which is triggered or exacerbated by spinal surgery in the same topographical location [9]. The clinical picture spans from mild axial pain to the most debilitating pain intangible to the underlying mechanism and refractory to medical management. Some authors consider this term a misnomer for both surgeons and patients alike as it represents a mismatch between preoperative expectations and the outcomes of surgery [10]. Throughout the years, many authors have struggled with the term FBSS. The main criticism was that this term is not only misleading but also potentially hurtful for practitioners as its initial pejorative connotation suggests failure or blame [11]. Many alternative terms have been proposed. In 2015, Rigoard et al. proposed the term Post Operative Persistent Syndrome (POPS) in turn [12]. The International Association for the Study of Pain (IASP) Task Force, which was formed in 2012 to work with members of the WHO to develop diagnostic codes for chronic pain, has recently undertaken a thorough revision of the classification of chronic pain within the ICD-11 classification [13]. The authors identified 13 serious issues with the term PSPS-T1/2 and suggested that impaired terminology is likely to hamper patients’ access to the best possible treatment. Upon discussion, workshop and Delphi panels the term Persistent Spinal Pain Syndrome (PSPS) was chosen to replace the former FBSS [14]. The stated aim of the Task Force was to present chronic pain in ICD-11 as a coherent category of diagnoses, unlike in ICD-10 where it was divided artificially [15]. Thus, it was not merely an attempt to come up with a better name but rather to systemize the largely heterogenic FBSS based on the (suspected) etiology. The authors integrated the PSPS into a previously formulated schema for spine-related pain. PSPS is the first level diagnosis, along with chronic secondary musculoskeletal pain, chronic neuropathic pain and chronic postsurgical/post-traumatic pain. On the next level, PSPS is split into type 1 (not associated with surgery) and type 2 (associated with surgery). Third-level diagnoses, associated with spondylosis and painful radiculopathy, can be parented by either type 1 or type 2 PSPS. The latter type can be a parent to chronic postsurgical or post-traumatic pain. The new nomenclature (PSPS-T1 and -T2) is now routinely used in publications [16,17].

It is important to note that PSPS-T1 is not synonymous with FBSS [18]. While both terms describe post-surgical spinal pain, PSPS-T1 specifically refers to cases where pain persists or worsens after spinal surgery due to the surgical intervention itself. In contrast, FBSS is a more general term that encompasses a variety of potential causes for persistent pain. Previously, as both PSPS Type 1 and Type 2 resulted in persistent pain post-surgery, they were classified under the umbrella term of FBSS. This labeling may have suggested that the surgery played a direct role in worsening the patient’s condition. However, with our growing knowledge and the shift towards the PSPS classification, this perspective has been clarified. We now understand that, while surgery may not yield the expected relief in some PSPS Type 1 cases, it does not necessarily imply that the surgery caused a worsening of the condition. In PSPS Type 2 cases, the persistent pain might be related to systemic conditions or the development of new conditions that are not directly tied to the surgical procedure itself. Thus, the transition to the PSPS terminology represents an advancement in our comprehension of post-surgical persistent pain. It moves away from the overly simplistic and potentially misleading notion that surgery ‘failed’ and instead acknowledges the complexity of these health issues, providing a more accurate and constructive framework for understanding and addressing them.

The distinction between these classifications aids in more targeted treatment and management strategies for patients experiencing post-surgical spinal pain, as FBSS is a general term used to describe the condition of patients who have not had a successful result from spinal surgery and continue to experience pain. This term does not provide specific information about the cause of the pain, and it may include various factors, such as surgical complications, inadequate decompression, or failure to address the primary cause of pain.

While PSPS T1 and T2 (PSPS-T1/2) do not exactly equate to FBSS, for the ease of comprehension in the subsequent text, they will be presented as the more contemporary and comprehensive replacements for the older term, FBSS.

## 3. Epidemiology

Estimating the prevalence of Failed Back Surgery Syndrome PSPS-T1/2 is challenging, but it can potentially affect as much as 20% of patients who have undergone spinal fusion procedures [19]. Lumbar surgeries, particularly spinal fusions, tend to have a higher risk of developing PSPS-T1/2 [20].

## 4. Molecular Mechanisms Underlying the Pathology of the PSPS-T1 (Surgery-Related)

PSPS-T1 is a complex and multifactorial condition that does not have a single unified pathology at the molecular level. However, several molecular mechanisms can contribute to the development or persistence of pain and other symptoms in PSPS-T1 patients. The surgical trauma can result in tissue damage and inflammation, which can cause the release of pain mediators such as cytokines, bradykinin, and prostaglandins [21]. This inflammatory response can cause ongoing pain and nerve irritation, even after the initial cause of the pain has been removed. In some cases, the development of scar tissue can be exacerbated by the use of certain surgical techniques or the use of surgical instruments that can damage surrounding tissue. Damage or irritation to the spinal nerves [22] during surgery can lead to alterations in the expression and function of ion channels, receptors, and neurotransmitters [23] in the affected nerves. This can result in abnormal pain signaling and the development of persistent neuropathic pain. Prolonged pain can lead to changes in the central nervous system, including the spinal cord and brain, resulting in central sensitization (i.e., generalized hypersensitivity) [24]. The epigenetic regulation of gene expression in the brain is essential for responding to enduring stress and chronic pain [25]. As a result, the pain threshold may be lowered, leading to heightened sensitivity to pain signals even after the original source of the pain has been addressed. Glial cells, which are non-neuronal cells in the nervous system, serve to maintain homeostasis and support neurons. However, in situations involving chronic pain, glial cells like microglia, astrocytes, and satellite glial cells undergo morphological and functional changes which contribute to increased pain sensitivity [26,27]. An additional significant factor to consider is the formation of scar tissue [28]. After spinal surgery, fibroblasts and other cells involved in wound healing can produce extracellular matrix proteins, leading to the formation of scar tissue [29]. This scar tissue can affect the local tissue environment and contribute to pain by compressing or tethering nerve roots [30]. It is essential to recognize that the molecular mechanisms contributing to PSPS-T1/2 can vary among individuals and may be influenced by factors such as the type of surgery, pre-existing medical conditions, and individual genetic differences. Understanding the specific molecular mechanisms involved in a patient’s PSPS-T1/2 can help guide targeted treatment strategies, such as anti-inflammatory medications, nerve modulation therapies, or other interventions designed to address the underlying molecular contributors to pain and dysfunction.

## 5. Etiology—Risk Factors

In the past, the concept of FBSS did not clearly distinguish between symptoms that were a direct result of surgery or those that arose due to the failure to resolve the main issue. Traditionally, predisposing factors were classified into three groups: preoperative, surgical, and postoperative. However, the updated classification system, which includes PSPS-T1 and PSPS-T2, more effectively emphasizes the impact of surgery on these outcomes.

### 5.1. Risk Factors for PSPS-T1

Various risk factors have been recognized for PSPS-T1, again, a condition characterized by pain that either continues or worsens following spinal surgery and is directly associated with the surgical procedure itself.

Surgical complications are one of the main concerns for surgeons and patients undergoing spinal surgery. Infection [31], neural damage [32], or hardware failure can occur during or after surgery and contribute to the development of PSPS-T1.

Incidental durotomy may result in persistent pain through various mechanisms, including direct injury to the nerves or inflammation [33]. Inadequate or inaccurate decompression during surgery can lead to the persistence or worsening of pain [34].

The formation of scar tissue surrounding the nerves is often referred to as epidural fibrosis. This phenomenon transpires when the body’s innate healing response prompts the formation of fibrous connective tissue within the epidural space [35]. Consequently, the nerves may experience compression or tethering, leading to discomfort and potentially playing a role in the genesis of PSPS-T1.

Intraoperative nerve or spinal cord injury during surgery can result in chronic pain and neurological deficits [36]. This type of pain frequently exhibits characteristics associated with neuropathic pain, making it a distinct and challenging condition to address. One of methods to prevent this is neurophysiological monitoring. Continuously assessing the functional integrity of neural structures can potentially help in reducing the risk of nerve or spinal cord injuries [37]. As a result, this may also have implications in reducing the incidence of postoperative neuropathic pain and its chronicity, thus playing a crucial role in improving patient outcomes.

Pseudoarthrosis in spinal fusion has been identified as a potential etiology for PSPS-T1 [38]. In cases where the intended spinal fusion does not achieve a solid bony union, the persistent motion and instability at the pseudoarthrosis site may contribute to the development of chronic pain and functional impairment. This form of non-union, by definition, refers to the lack of complete bony fusion observed at least six months following a surgical procedure [39]. The manifestation of pseudoarthrosis may vary depending on individual factors and the nature of the surgery. Generally, symptoms associated with pseudoarthrosis can begin to appear within a few months to a year postoperatively. However, the precise timeline may differ for each patient, and, in some cases, it may take longer for the symptoms to become evident.

Understanding and managing these risk factors before and after spinal surgery can help reduce the likelihood of developing PSPS-T1 and improve patient outcomes. Proper surgical planning, technique, and postoperative care play crucial roles in minimizing the risks associated with PSPS-T1.

### 5.2. Risk Factors for PSPS-T2

The risk factors for Persistent Spinal Pain Syndrome Type 2 (PSPS-T2) can vary depending on the individual case and the primary cause of the pain. However, there are some common risk factors for developing PSPS-T2.

Incorrect diagnosis is arguably a leading factor [20,40]. If the primary cause of the pain was not accurately identified prior to surgery, the procedure may fail to address the underlying issue, leading to persistent or worsening pain (Figure 1). Foraminal stenosis is often a subject of misdiagnosis [41,42]. The ensuing compression of nerve roots can manifest as pain, paresthesia, and weakness in the corresponding dermatomes. Owing to the symptomatic overlap with other spinal disorders, the accurate diagnosis of foraminal stenosis demands meticulous clinical assessment and the employment of appropriate imaging modalities to facilitate optimal treatment and management strategies. Even with the correct diagnosis, if the surgical technique or approach is not appropriate for the patient’s specific condition, the surgery may not be successful in resolving the pain. In cases of spinal stenosis, surgeons may inadvertently overestimate the extent of decompression achieved during surgery, potentially leading to inadequate decompression [43,44]. This suboptimal outcome can contribute to persistent symptoms and complications, emphasizing the need for a thorough intraoperative assessment to ensure the appropriate level of decompression is attained.

New-onset conditions may follow surgery, such as adjacent segment degeneration [45], recurrent disc herniation [46], or scar tissue formation (epidural fibrosis) [47]; these can emerge after spinal surgery and contribute to the development of PSPS-T2.

Psychological factors play a crucial role in the perception and persistence of pain following surgery. Various emotional factors, including depression, anxiety, and unrealistic expectations regarding surgical outcomes [48,49,50], can contribute to the continued experience of pain after the procedure.

Systemic medical comorbidities like diabetes, obesity, or osteoporosis can negatively impact healing and recovery after spinal surgery [51,52,53], increasing the risk of PSPS-T2.

Another concern frequently brought up is the comparatively limited efficacy of spinal surgeries in general. In recent years, spinal surgery has come under increased scrutiny due to a variety of factors. Among these, the most notable is the relatively low success rate, particularly when contrasted with outcomes in other medical specialties. In 2013, Mannion et al. published a significant study comparing the effectiveness of spinal surgery to that of large joint replacements [54]. This study offers a comprehensive evaluation of the relative success of surgical interventions for degenerative disorders of the lumbar spine, hip, or knee in over 4500 patients. The results revealed that spinal surgery had a considerably lower success rate when compared to hip and knee surgery. Specifically, lumbar decompression surgery had a 67% success rate, lumbar fusion surgery had a mere 55% success rate, whereas hip and knee replacements demonstrated success rates of 89% and 81%, respectively. This research emphasizes the importance of conducting an honest evaluation of surgical outcomes to enhance patient care and facilitate informed decision-making. In accordance with the updated nomenclature, up to 45% of patients may fall into either the PSPS-T1 or PSPS-T2 categories.

## 6. Diagnosis

### 6.1. History

History taking is a critical component of the patient assessment process, as it allows healthcare professionals to gather essential information about the patient’s symptoms, medical history, and psychosocial factors [22].

When evaluating patients with PSPS T1/T2, employing the concept of red and yellow flags can be a convenient and effective approach. Red flags are potential indicators of a serious underlying condition that requires immediate attention or further investigation. Identifying red flags can help guide appropriate diagnostic testing and management, and ensure that potentially life-threatening or disabling conditions are not overlooked. Common red flags in the context of back pain are listed in Table 1.

Yellow flags, on the other hand, are psychosocial factors that may increase the risk of chronicity or disability in patients with back pain [55]. Identifying and addressing yellow flags can help guide appropriate referrals for psychological support and improve the overall management of the patient’s pain. Common yellow flags are also shown in Table 1.

**Table 1 medicina-59-01255-t001:** Identification of red and yellow flags in low back pain: symptoms, signs, and factors indicating potential serious conditions and inhibitors to recovery.

Symptoms and Signs	Explanation
Red Flags [4,5]	
Recent Trauma	Onset under 20 or over 50 years
History of Cancer	Any trauma, minor or major, especially in people over 50
Unexplained Weight Loss	Especially if new back pain in patients with known cancer history
Failure to Improve	Can indicate cancer or systemic disease
Night Pain	Pain not improving after 4–6 weeks of appropriate conservative treatment
Fever, Chills, Sweats	Pain that wakes you up at night
Pain Not Relieved by Rest or Lying Down	May indicate infection or systemic disease
History of Intravenous Drug Use	Often indicates a serious condition
History of Long-Term Steroid Use	Increased risk of infection
Neurologic Symptoms	Increased risk of osteoporosis and vertebral fractures
Bowel or Bladder Dysfunction	Such as weakness, numbness, or altered sensation in lower extremities
Severe or Progressive Neurological Deficit	Could indicate cauda equina syndrome, a surgical emergency
Recent Trauma	Indicates possible nerve involvement and requires urgent attention
Yellow Flags [56]	
Fear-Avoidance Behavior	Avoiding movement due to fear of causing more pain
Belief That Pain Means Harm	Incorrectly associating all pain with harm can inhibit recovery
Catastrophizing	An exaggerated negative view of the pain’s impact
Expectation of Passive Treatments Only	Belief that only treatments performed on the person (like surgery, injections) will help, rather than active participation
Depression, Anxiety, or Stress	Psychological factors can influence the perception of pain and recovery
Over-reliance on Medication	May indicate lack of active coping strategies
Poor Job Satisfaction or Difficulties at Work	Could influence chronicity and disability

During history taking, it is essential to thoroughly explore both red and yellow flags to ensure a comprehensive understanding of the patient’s condition. This information will help guide the diagnostic workup, treatment plan, and any necessary referrals, ultimately optimizing patient outcomes and reducing the risk of chronic pain or disability.

### 6.2. Imaging

The use of diagnostic imaging methods is pivotal in identifying the root cause of PSPS-T1/2 and guiding the course of future treatment plans. A variety of imaging techniques can be employed to assess patients suffering from this condition.

Radiography (X-ray): Radiographic imaging is a fundamental and non-invasive technique to evaluate spinal alignment, osteophytes, and the positioning of surgical hardware (e.g., screws, rods, or cages). Assessing spinopelvic alignment is vital for determining causes and planning corrective measures, especially after fusion procedures and in treating spinal deformities [57,58]. Sagittal misalignment can contribute to both PSPS-T1 and PSPS-T2. In the case of PSPS-T1, improper alignment correction might result in suboptimal outcomes. For PSPS-T2, attempting to correct an already inappropriate alignment could potentially exacerbate the issue and lead to further complications. Crucial spinopelvic parameters include pelvic incidence (PI), pelvic tilt (PT), sacral slope (SS), lumbar lordosis (LL), and thoracic kyphosis (TK). PI-LL mismatch, where lumbar lordosis deviates significantly from pelvic incidence, can lead to abnormal spinal alignment, biomechanics, and PSPS-T1/2 development [59,60,61]. Overcorrection and under-correction during surgery can also result in postoperative pain [62,63]. Overcorrection may cause stress on adjacent spinal segments, leading to adjacent segment disease, while under-correction can cause continued pain, nerve compression, and abnormal spinal biomechanics, contributing to PSPS-T1/2. Evaluating spinopelvic parameters and their interrelationships is essential for identifying PSPS-T1/2 causes and planning suitable surgical interventions. However, radiography is limited in its capacity to visualize soft tissues such as muscles, ligaments, and intervertebral discs.

Magnetic Resonance Imaging (MRI): MRI is a valuable imaging modality for examining soft tissues, such as the spinal cord, nerve roots, and intervertebral discs. It can identify disc herniation, spinal stenosis, scar tissue, nerve root compression, and other potential causes of both types of PSPS [64,65]. Traditionally, MRI was limited due to the presence of metallic implants causing artifacts, which reduced image quality and diagnostic usefulness. However, new techniques now enable a reduction in these artifacts, improving diagnostic capabilities [66,67]. MRI variations, such as MR neurography of the lumbosacral plexus, can increase the sensitivity even further in identifying issues associated with PSPS-T1/2 [68].

Computed Tomography (CT): CT scans provide detailed cross-sectional images of bony structures, which can be helpful in assessing spinal fusion [69], hardware complications [70], and to assess the adequacy of decompression [71]. Other adaptations of CT scans further enhance the diagnostic capabilities in identifying the causes of PSPS-T1/2. SPECT-CT (Single Photon Emission Computed Tomography-Computed Tomography), a variation of CT, serves as a diagnostic instrument capable of identifying pain sources in patients with chronic neck and back pain [72]. This results in improved specificity and contributes to more effective pain management for patients.

Myelo-CT: Myelo-CT, also known as CT myelography, is a diagnostic imaging technique that combines computed tomography (CT) with myelography. This procedure involves the injection of a contrast agent into the cerebrospinal fluid space surrounding the spinal cord and nerve roots. The CT scan is then performed to obtain detailed images of the contrast-filled spinal canal and nerve roots. Myelo-CT can be employed to evaluate spinal canal narrowing [73], nerve root compression, and the presence of scar tissue [74]. Additionally, myelo-CT serves as a valuable alternative in cases where the presence of metallic implants preclude or limit the use of MRI, as the implants may cause artifacts that compromise image quality and diagnostic utility [75]. Although myelo-CT is more invasive than other imaging modalities, such as MRI, it may be particularly useful in cases where MRI is contraindicated or inconclusive.

Positron Emission Tomography (PET): PET scans can be employed to identify areas of inflammation, infection, or abnormal metabolism in the spine [76,77]. While not routinely used for PSPS-T1/2, PET scans can provide supplementary information when other imaging modalities are inconclusive.

Epidurography is a diagnostic imaging technique that involves the injection of a contrast agent into the epidural space of the spine. This procedure is primarily used to visualize the anatomy of the epidural space, including the spinal nerves and surrounding structures. It is particularly beneficial in detecting abnormalities like adhesions, scar tissue, or epidural fibrosis, which could be responsible for ongoing pain [78,79]. Epidurography is often performed in conjunction with an epidural steroid injection [80], which is a therapeutic intervention aimed at reducing inflammation and alleviating pain. The real-time visualization provided by the procedure allows the physician to guide the needle and ensure the accurate placement of the steroid medication [81]. While epidurography can provide valuable information about the epidural space, it has some limitations. It is an invasive procedure and carries a risk of complications such as infection, bleeding, or nerve damage [82]. Additionally, epidurography does not provide the same level of detail as other imaging modalities, such as MRI or CT, for assessing the soft tissue structures and bony anatomy of the spine [83]. As a result, epidurography is typically used as an adjunct to other imaging techniques or when other modalities are contraindicated or inconclusive.

### 6.3. Other

In addition to imaging techniques, other diagnostic methods, such as electromyography (EMG) [83] and diagnostic injections, can be employed to evaluate patients with back pain or PSPS-T1/2. These methods can provide valuable information about the underlying causes of pain and guide treatment decisions. EMG can help identify nerve compression, irritation, or damage, which may contribute to a patient’s pain. EMG is often performed in conjunction with nerve conduction studies (NCS) to provide a more comprehensive assessment of the peripheral nervous system. EMG and NCS can be particularly useful in differentiating between nerve root compression (e.g., radiculopathy) and other peripheral nerve disorders (e.g., neuropathy) [84,85].

Diagnostic injections are invaluable in accurately locating the origin of a patient’s pain in PSPS-T1/2 by focusing on particular nerves or structures. These injections typically involve the administration of a local anesthetic, with or without a corticosteroid, to provide temporary pain relief and diagnostic information. There are several types depending on the targeted structure. Facet joint injections target the facet joints that can be implicated in the genesis of pain [86,87]. The substantial alleviation of pain post-injection could signify the facet joints as the pain source. Medial branch blocks are also aimed at the facet joints to identify them as potential origins of pain. They are executed by anesthetizing the medial branch nerves responsible for facilitating sensation to the facet joints. These blocks also aid in discerning if the facet joints are, indeed, the culprits behind the pain [88,89]. Selective nerve root blocks, on the other hand, target precise spinal nerve roots, thus aiding in identifying nerve root compression or irritation as potential etiologies for the patient’s pain manifestation. Epidural injections, typically utilized for therapeutic interventions, can concurrently serve a diagnostic purpose. If significant pain relief follows an epidural injection, it could be indicative of spinal nerve inflammation or compression as a contributory factor to the patient’s pain [86,90].

When evaluating patients with back pain or PSPS-T1/2, a combination of diagnostic methods, including imaging, EMG, and diagnostic injections, may be necessary to accurately identify the underlying cause of pain and guide treatment decisions.

## 7. Management

### 7.1. Causative Treatment

Interventions intended to treat the root cause of a patient’s pain are preferred. One such intervention is surgical treatment, which is frequently the most rational first-line approach. When conservative management strategies do not provide relief, or there is a definitive structural origin for the pain, surgical procedures, such as spinal stabilization, fusion, or decompressive procedures, may be indicated [84,85].

Targeted physical therapy constitutes a critical component in the continuum of treatment. It can help address muscle imbalances, improve flexibility, and increase core strength, all of which can contribute to reducing pain and preventing future episodes [86,87]. Advanced physiotherapeutic interventions, including targeted exercises and stretching regimens, in conjunction with precise manual therapy methodologies, can be customized to suit the specific requirements and condition of the patient [88]. In addition, chiropractic interventions and osteopathic manipulations can contribute significantly to managing Failed Back Surgery Syndrome (PSPS-T1/2). These therapeutic approaches primarily address the alignment and functionality of the musculoskeletal system, especially the spinal column, assisting in the mitigation of specific pain types and enhancement of functional capacity [86,89]. These therapies can be integrated with other strategies, such as pharmacological treatments, physical therapy, and lifestyle modifications. However, it is important to underscore that these therapies may not be a universal solution, as they may not be appropriate for all patients. In particular, individuals presenting with certain forms of spinal instability or specific neurological manifestations may not be suitable candidates for these treatments.

Injections, already described above as being part of the diagnostic process, can also function in a therapeutic capacity, serving as a causative treatment that directly targets and mitigates the origin of pain [90]. Epidural steroid injections and/or nerve blocks can be used to target specific sources of pain, and reduce inflammation [91], in addition to the already mentioned diagnostic information. Alternative forms of injection therapies may also be explored for treatment purposes. Prolotherapy, a therapeutic approach involving injections to trigger healing and alleviate pain, appears to offer promising possibilities in the treatment of PSPS-T1/2. A recent study by Solmaz et al. [92] demonstrated the potential effectiveness of 5% dextrose injections for treating Failed Back Surgery Syndrome (PSPS-T1/2). Their research found significant improvements in pain and functionality, thereby suggesting this as a promising alternative before resorting to revision surgery.

### 7.2. Symptomatic Treatment

Symptomatic treatments focus on managing pain and improving the patient’s quality of life and medications often represent the first line of intervention for treatment across the board. The choice of medication typically depends on the nature of the pain, whether it is neuropathic or nociceptive, and its severity.

Nociceptive pain, resulting from tissue damage or inflammation, may first be treated with over-the-counter (OTC) non-steroidal anti-inflammatory drugs (NSAIDs) like ibuprofen and naproxen or acetaminophen [93]. If these are ineffective, prescription non-opioid medications, like higher-dose NSAIDs, COX-2 inhibitors, or short-term corticosteroids, may be introduced [94]. Neuropathic pain, stemming from nerve damage or dysfunction, may initially respond to drugs like gabapentin or pregabalin, specifically designed to treat this type of pain [93]. Certain antidepressants, like duloxetine or amitriptyline, can also be effective in managing neuropathic pain [95]. Radicular pain, a specific type of neuropathic pain following the path of a nerve root from the spine, can also be treated with neuropathic pain medications, along with physical therapy and possibly corticosteroid injections for more direct relief [96,97]. When it comes to the use of systemic steroids, these have shown effectiveness in the acute management of sciatica [98]. However, their application in the long-term treatment of chronic pain conditions, including Failed Back Surgery Syndrome (PSPS-T1/2), is typically constrained owing to the associated side effects [99]. Axial pain, which is localized in one area of the spine, may respond to NSAIDs, physical therapy, and, in some cases, to corticosteroid injections or nerve blocks. For severe pain unresponsive to other treatments, opioids such as tramadol, morphine, oxycodone, or hydrocodone may be considered [96]. Specialized treatments, like implantable drug delivery systems, may also be employed [100]. This tiered approach should be customized to each patient’s needs and overseen by a healthcare provider, taking into account the potential side effects and risks associated with each treatment. Patient’s responses to these treatments can vary greatly, and it often takes time to find the right medication or combination of medications that will provide the best balance between pain relief and side effects. The broad overview of pharmacological treatments is presented in Table 2.

**Table 2 medicina-59-01255-t002:** A summary of the primary pharmacological approaches in the management of PSPS-T1/T2.

Name of Drug	Main Indication in PSPS-T1/2	Potential Side Effects
Tier 1 (Non-opioid analgesics)		
Acetaminophen (Paracetamol) [101]	Mild to moderate axial pain	Liver damage, skin reactions, kidney damage
Non-Steroidal Anti-Inflammatory Drugs (NSAIDs), e.g., Ibuprofen, Naproxen [101],	Axial pain, sciatica/radicular pain	Stomach ulcers/bleeds, increased risk of heart attack or stroke
Acetaminophen (Paracetamol) [101]	Mild to moderate axial pain	Liver damage, skin reactions, kidney damage
Tier 2 (Mild opioids for moderate to severe pain)		
Codeine [42] (often combined with Acetaminophen)	Moderate to severe axial pain	Drowsiness, constipation, nausea
Tramadol [42]	Moderate to severe axial pain	Nausea, dizziness, constipation, risk of addiction
Tier 3 (Strong opioids for severe pain)		
Morphine [42]	Severe axial pain	Drowsiness, constipation, nausea, risk of addiction
Fentanyl [42]	Severe axial pain	Drowsiness, constipation, nausea, risk of addiction
Adjuvant analgesics (medications that can enhance pain relief or combat side effects)		
Anticonvulsants—Gabapentin [102], Pregabalin [103]	Neuropathic pain related to nerve damage in PSPS-T1/2	Dizziness, fatigue, weight gain
Antidepressants—Amitriptyline, Duloxetine [104]	Neuropathic pain and associated depressive symptoms in PSPS-T1/2	Drowsiness, dry mouth, constipation, weight gain
Anticonvulsants—Gabapentin [102], Pregabalin [103]	Neuropathic pain related to nerve damage in PSPS-T1/2	Dizziness, fatigue, weight gain

One treatment with proven efficacy in managing PSPS, especially type 1, is neurostimulation [105]. It is usually viewed as the ultimate recourse when all other causative treatments have proven ineffective, or when symptomatic treatment does not provide adequate pain alleviation. Utilizing electrical impulses to disrupt pain signaling pathways, it provides substantial relief from the persistent pain synonymous with PSPS-T1, thereby enhancing patients’ functional capacity and overall quality of life. Spinal cord stimulation (SCS), in particular, has demonstrated notable efficacy and versatility among the array of neurostimulation techniques. Despite not addressing the underlying PSPS causality, the merits of this symptomatic management approach are increasingly recognized within the neurosurgical community [106]. However, given the variability in individual responses, it remains critical to devise patient-specific treatment plans to maximize the therapeutic benefits of neurostimulation [107]. There are several neurostimulation therapies other than SCS that have been investigated for the treatment of PSPS-T1/2. However, the evidence for their effectiveness is limited. One such therapy is dorsal root ganglion (DRG) stimulation, which has shown promising results in small studies [108]. Another therapy is peripheral nerve stimulation (PNS), which has been used to treat PSPS-T1/2 with mixed results [109]. Additionally, motor cortex stimulation (MCS) has been investigated as a treatment option, but the evidence for its effectiveness is limited [110]. Overall, while there are alternative neurostimulation therapies to SCS for the treatment of PSPS-T1/2, more research is needed to determine their effectiveness.

In the comprehensive treatment approach to effective treatment, the role of psychological support should not be underestimated. Living with chronic pain can impose significant psychological distress, with patients often experiencing anxiety, depression, and feelings of helplessness [111]. This psychological burden can, in turn, intensify the perception of pain, thereby perpetuating a detrimental cycle [112]. Integrating psychological support in the form of interventions such as Cognitive Behavioral Therapy (CBT) can empower patients with essential skills to manage these psychological aspects effectively [85]. CBT, in particular, can help patients recognize and reframe negative thought patterns, develop healthier coping mechanisms, and employ relaxation techniques [113]. Consequently, this can lead to improved pain management, enhanced mental well-being, and overall improved quality of life [87]. Implementing psychological support as an integral part of PSPS-T1/2 treatment underscores the importance of a holistic, patient-centered approach to care. This strategy, in combination with appropriate medical and physical therapies, can substantially contribute to more favorable outcomes in PSPS-T1/2 management.

A comprehensive treatment plan for PSPS-T1/2 or back pain should include a combination of causative and symptomatic treatments, tailored to the specific needs and circumstances of each patient. Multidisciplinary approaches that involve input from various healthcare professionals (e.g., physicians, physical therapists, psychologists) are often most effective in managing complex pain conditions.

## 8. Conclusions

PSPS-T1/2, formerly known as FBSS, is a multifaceted and complex issue. The wide range of pain types and diverse patient outcomes highlight the fact that this entity is not a one-size-fits-all diagnosis, but rather a spectrum of possible conditions. The root cause of the pain can vary widely among patients and may be linked to many factors, including surgical technique, the original diagnosis, patient health, and even psychosocial factors.

The various manifestations of PSPS-T1/2 demonstrate the complexity and heterogeneity of this condition. It may present as mild, axial pain, or escalate to debilitating pain resistant to medical management. This complexity makes PSPS-T1/2 a significant challenge for healthcare providers. Identifying the cause of the pain after surgery and subsequently providing effective treatment becomes increasingly difficult due to the broad range of potential underlying mechanisms.

The introduction of the new nomenclature, Persistent Spinal Pain Syndrome (PSPS), and its two types (PSPS-T1 and PSPS-T2), represents an important step toward a more nuanced understanding of these complex conditions. The terms PSPS-T1 and PSPS-T2 not only remove the stigmatizing implication of failure associated with PSPS-T1/2, but also encourage a more systematic approach to diagnosing and discussing post-surgical spinal pain. This new classification approach aims to take into account the complexity of the condition and improve both the diagnosis and treatment strategies, ultimately enhancing patient outcomes.

## Figures and Tables

**Figure 1 medicina-59-01255-f001:**
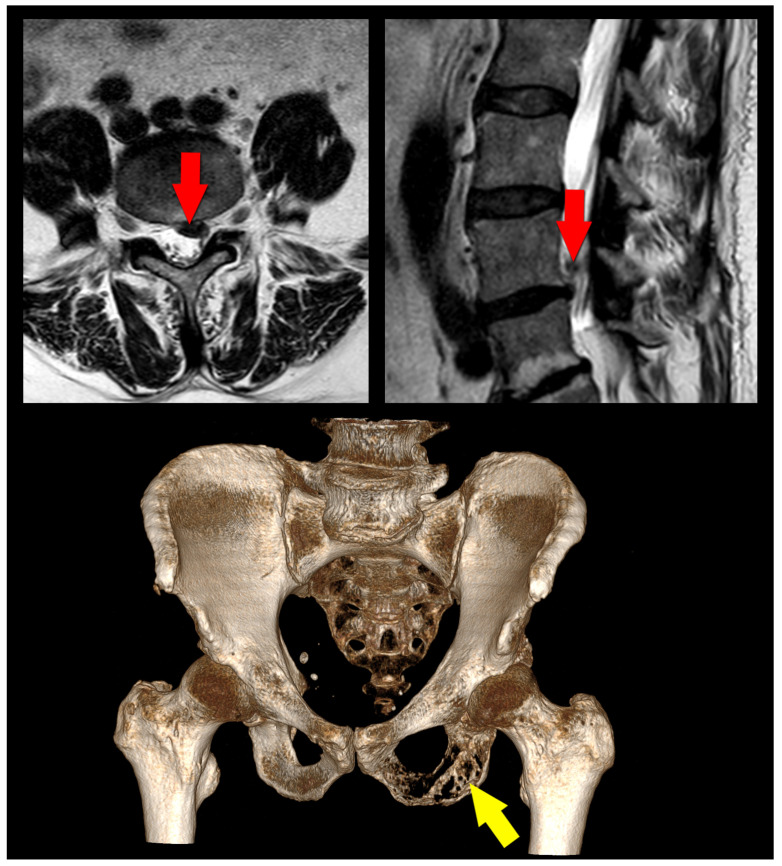
A case illustration for PSPS-T2. A 62-year-old patient presented with a 2-month history of left-sided sciatica that had been progressively worsening. The patient was neurologically intact and primarily experienced radicular pain consistent with the L5 dermatome. Magnetic resonance imaging (MRI–top row) revealed a disc herniation at the L4/5 level with compression of the L5 nerve root inside the spinal canal (red arrows). The patient received a transforaminal epidural steroid injection, but it only offered temporary pain relief for three days. Ultimately, a decision was made to move forward with the surgical intervention. The patient underwent an L4/5 microdiscectomy and initially made a good recovery. However, the pain returned within a week, and a new type of pain, primarily located in the left buttock, developed. It started to have a worrying nociceptive component, as evidenced by the presence of allodynia and hyperesthesia. Furthermore, the pain intensity was heightened during nighttime hours and remained unaffected by changes in body position. The patient soon became unable to sit on his left buttock. As the pain grew more intense and was distinctly concentrated in the ischial tuberosity, further examinations were conducted. A computed tomography (CT) scan revealed extensive bone destruction in the left ischial bone (yellow arrow), while a subsequent MRI scan indicated the presence of a likely malignant tumor compressing the sciatic nerve near the ischial tuberosity (not shown). Additional investigations eventually diagnosed the patient with advanced-stage lung cancer. Without delay, the patient was referred to a specialized institution for interdisciplinary treatment.

## Data Availability

No new data were created.

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
