# Peer review of "Failed Back Surgery Syndrome: No Longer a Surgeon’s Defeat—A Narrative Review"

_medicina, 2023, doi:10.3390/medicina59071255_

Round 1

Reviewer 1 Report

  1. Can Persistent Spinal Pain Syndrome (PSPS-T1/2), replace the  older term Failed Back Surgery Syndrome (FBSS) ?
  2. Yes it is original, the suggestion that all pain after surgery is not related to Failed Back Surgery Syndrome is right.
  3. The term Persistent Spinal Pain Syndrome is added by that manuscript.
  4. I do not recommend further controls.
  5. The conclusions are consistent.
  6. The references are apropriate.
  7. No additional comments.

Author Response

Dear reviewer,

I would like to express my sincere gratitude for your positive feedback and thorough review of my manuscript. I appreciate the time and effort you have dedicated to the review process.

In terms of q1: yes, my article addresses exactly this topic; it elaborates on how the term Persistent Spinal Pain Syndrome (PSPS-T1/2) has been introduced to replace the older term Failed Back Surgery Syndrome (FBSS). The article discusses the reasons for this change in nomenclature, the implications it holds for patient diagnosis and treatment, and the potential benefits of adopting the new term.

I acknowledge your comment regarding the absence of the need for further controls and your satisfaction with the consistency of the conclusions. It is encouraging to receive your affirmation that the references are appropriate.

As you did not have any additional comments or suggestions for revisions, I will take the positive nature of your review as an indication that the manuscript is ready for publication in its current form.

Once again, thank you for your valuable contribution.

Kind Regard,

G. Miekisiak

Reviewer 2 Report

Well-written article, with extensive literature review and low back pain mechanisms. Undoubtedly the introduction of the new nomenclature, Persistent Spinal Pain Syndrome (SPPS) and its two types (PSPS-T1 and PSPS-T2), not only removes the stigma of surgical failure and helps to improve both diagnosis and treatment strategies.
Considerations:
Figure 1  “A case illustration  for PSSP-T1” Exemplifies PSSP-T1, but is not mentioned in the text. Consideration should be given to the image in the text, and not just in the description.

“Intraoperative nerve or spinal cord injury during surgery can result in chronic pain and neurological deficits. This type of pain frequently exhibits characteristics associated with neuropathic pain, making it a distinct and challenging condition to address”.

Intraoperatively, we can perform neurophysiological monitoring to minimize neurological damage. Could monitoring interfere with postoperative neuropathic pain and chronicity? This information is worth adding.

Author Response

Thank you very much for your positive feedback on the article and for acknowledging the extensive literature review and discussion on low back pain mechanisms. I'm glad you recognize the significance of the new nomenclature, Persistent Spinal Pain Syndrome (PSPS), and its types (PSPS-T1 and PSPS-T2) in removing the stigma and improving diagnosis and treatment strategies.

Regarding your considerations:

  1. Figure 1 - “A case illustration for PSSP-T1”: You have rightly pointed out that Figure 1 is not explicitly mentioned in the text. Additionally, there was an error in the labeling of the figure; it should be PSPS-T2, not PSPS-T1. This has been corrected, and I have also ensured that there is a reference to this figure in the text to provide context.

  2. Suggestion on Intraoperative Neurophysiological Monitoring: Thank you for bringing up the importance of intraoperative neurophysiological monitoring. It's an excellent point, and its potential role in minimizing neurological damage and possibly impacting postoperative neuropathic pain and chronicity is valuable information. I have added a relevant section.

Your insightful comments are greatly appreciated and have contributed to the improvement of the manuscript.

Kind Regard,

G. Miekisiak

Reviewer 3 Report

This is an interesting and well-structured narrative review. There are some suggestions that should be considered:

1.Please re-check the numeric order of the headings and sub-headings, e.g. heading 3 was repeated more than once.

2.I suggest generating a decision tree or a diagram to show management algorithm of PSPS. This will help readers understand the key points better.

Author Response

Dear Reviewer,

Thank you for your valuable feedback and positive remarks on my narrative review. I appreciate your time and insights.

  1. I have taken note of the issue regarding the numeric order of the headings and sub-headings and have corrected the numbering sequence to ensure clarity.

  2. Your suggestion of generating a decision tree or diagram to aid readers in understanding the management algorithm of PSPS is spot-on. I would like to inform you that after the initial submission, I added a graphical abstract which includes a detailed illustration outlining the management algorithm of PSPS (see below). This addition will be displayed along the text abstract, and it aligns with your suggestion and will greatly assist readers in comprehending the key points.

Thank you once again for your constructive comments. Your input has been instrumental in enhancing the quality of my manuscript.

Kind Regard,

G. Miekisiak

Reviewer 4 Report

I commend the author for their research entitled “Failed Back Surgery Syndrome: No Longer a Surgeon's Defeat - A Narrative Review.” In his study the author focused on the influence of the introduction of the new term Persistent Spinal Pain Syndrome replacing the older term Failed Back Surgery Syndrome on the approach to diagnosing and treating post-surgical spinal pain.

Generally, the topic is very interesting, the manuscript is well written, the illustrative case is appropriate, the conclusion is based on results, and the literature is comprehensive and updated.

Specifically, in the Etiology section (lines 161 and 162) the author stated: “Infection [31], neural damage [32], or hardware malfunction can occur during or after surgery and contribute to the development of PSPS-T1.” What is meant by hardware malfunction? Was the hardware not chosen appropriately? There are still surgeons who are using “dynamic” spinal stabilization (please see and comment:  Fokter SK, Strahovnik A. Dynamic versus rigid stabilization for the treatment of disc degeneration in the lumbar spine. Evid Based Spine Care J. 2011 Aug;2(3):25-31. doi: 10.1055/s-0030-1267110.)

Author Response

Dear Reviewer,

I am deeply appreciative of your commendation and positive feedback regarding my manuscript. I am thrilled that you found the topic engaging.

I would like to address the specific comment you made concerning the use of the term "hardware malfunction" in the Etiology section. Upon reflection, I understand that the word "malfunction" may have been unfortunate. The more accurate term is "hardware failure", to encompass issues such as breakage, loosening, or migration of the implants often mentioned in the context of PJK/PJF. My intention was to underline the complications associated with the implanted hardware that could contribute to the development of PSPS-T1. The manuscript has been adjusted accordingly.

Thank you once again for your thoughtful and constructive comments which have significantly contributed to the refinement of my manuscript.

Warm regards,

G. Miekisiak